# Development of a Genome-Informed Protocol for Detection of *Pseudomonas amygdali* pv. *morsprunorum* Using LAMP and PCR

**DOI:** 10.3390/plants12244119

**Published:** 2023-12-10

**Authors:** Daniela Díaz, Alan Zamorano, Héctor García, Cecilia Ramos, Weier Cui, Claudia Carreras, María Francisca Beltrán, Boris Sagredo, Manuel Pinto, Nicola Fiore

**Affiliations:** 1Laboratorio de Fitovirología, Departamento de Sanidad Vegetal, Facultad de Ciencias Agropecuarias, Universidad de Chile, Avenida Santa Rosa 11315, Santiago 8820808, Chile; daniela.diaz.l@ug.uchile.cl (D.D.); agezac@uchile.cl (A.Z.); cuiweierpku@gmail.com (W.C.); claudia.carreras@gmail.com (C.C.); 2Laboratorio Diagnofruit, Avenida Sucre 1521, Santiago 7770273, Chile; hgarcia@diagnofruit.cl (H.G.); cecibrb@gmail.com (C.R.); 3Núcleo de Investigaciones Aplicadas en Ciencias Veterinarias y Agronómicas, Facultad de Medicina Veterinaria y Agronomía, Universidad de las Américas, Campus Providencia, Manuel Montt 948, Santiago 7500975, Chile; 4Instituto de Investigaciones Agropecuarias, INIA Rayentué, Avda. Salamanca s/n, Rengo 2940000, Chile; fran.ibv24@gmail.com (M.F.B.); bsagredo@inia.cl (B.S.); 5Instituto de Ciencias Agroalimentarias Animales y Ambientales (ICA3), Universidad de O’Higgins, Campus Colchagua, Ruta I-90 S/N, San Fernando 3072590, Chile; manuel.pinto@uoh.cl

**Keywords:** Prunus avium, LAMP, Pseudomonas syringae pv. morsprunorum, genome sequencing

## Abstract

One of the causal agents of bacterial canker is *Pseudomonas amygdali* pv. *morsprunorum*—Pam (formerly *Pseudomonas syringae* pv. *morsprunorum*). Recently detected in Chile, Pam is known to cause lesions in the aerial parts of the plant, followed by more severe symptoms such as cankers and gummosis in the later stages of the disease. This study presents the design of PCR and LAMP detection methods for the specific and sensitive identification of *Pseudomonas amygdali* pv. *morsprunorum* (Pam) from cherry trees. Twelve *Pseudomonas* isolates were collected, sequenced, and later characterized by Multi-locus Sequence Analysis (MLSA) and Average Nucleotide Identity by blast (ANIb). Three of them (11116B2, S1 Pam, and S2 Pam) were identified as *Pseudomonas amygdali* pv. *morsprunorum* and were used to find specific genes through RAST server, by comparing their genome with that of other *Pseudomonas*, including isolates from other Pam strains. The effector gene *HopAU1* was selected for the design of primers to be used for both techniques, evaluating sensitivity and specificity, and the ability to detect Pam directly from plant tissues. While the PCR detection limit was 100 pg of purified bacterial DNA per reaction, the LAMP assays were able to detect up to 1 fg of purified DNA per reaction. Similar results were observed using plant tissues, LAMP being more sensitive than PCR, including when using DNA extracted from infected plant tissues. Both detection methods were tested in the presence of 30 other bacterial genera, with LAMP being more sensitive than PCR.

## 1. Introduction

Bacterial canker in cherry trees is rapidly spreading in Chile due to the exponential expansion of the crop during the last decade. The disease is caused by different species from *Pseudomonas* genus, being *Pseudomonas syringae* pv. *morsprunorum* (now known as a *Pseudomonas amygdali* pv. *morsprunorum* (Pam), name which we will refer in this study) one of the most aggressive species [1,2,3,4]. Pam was first detected in Chile in 2019, in an orchard in Osorno city, Región de Los Lagos [5]. Even when Pam is not widely spread in Chile, it is urgent to develop fast and reliable detection methods to minimize the dissemination and future damage in cherry plants due to the presence of this bacteria, considering the relevance of cherry industry for the country.

The disease caused by *Pseudomonas amygdali* pv. *morsprunorum* shares a significant number of epidemiological aspects with *Pseudomonas syringae* pv. *syringae* (Pss). Commonly, the symptoms associated with bacterial canker are lesions that turns from brown to black in all aerial parts of the plant, such as flowers, fruits, and especially leaves and other green tissues, which are likely to become smooth after a few days [6]. Pam remains on the surface of woody dormant tissue during the winter in an epiphytic manner, and when spring arrives, enters to the plant through wounds or natural surface opening structures where it starts to actively multiplicate, colonizing a major area and causing the symptoms described above [7]. If the bacteria remain over time, it may become a systemic disease, causing cankers on woody tissue. Pam also can be associated with other species of *Prunus* and it is reported as one of the most aggressive species in the genus, especially associated with cherry trees [8].

The relationship between the different species in the genus *Pseudomonas* was defined by phylogenetic studies using Average Nucleotide Identity by Blast (ANIb) and Multilocus Sequence Analysis (MLSA). Pam was identified as a member of phylogroup IV, distinct from the *Pseudomonas syringae* phylogroup, which includes Pss, another species related to bacterial canker [9]. This conclusion is supported by several studies that validate the use of one or both methods of genetic comparison for taxonomic and phylogenetic studies [8,9,10,11,12]. Recently, it has been shown that the use of three or four housekeeping genes could clarify the phylogenetic relationships in *Pseudomonas* species [13,14], although it is known that the composition of the different phylogroups can vary depending on the type and number of genes chosen [15]. On the other hand, ANIb makes it possible to determine whether two or more bacterial strains belong to the same species when the percentage of genome similarity is 95%, considering a much wider range of genes. In this regard, strains such as Pam are clearly distinct from other *Pseudomonas* associated with bacterial canker and provide a point of comparison for new phylogenetic and genomic analyses. 

The selection of target genes for primer design is an important step in the development of a detection method. In this regard, pathogenicity-related genes are of great interest because they may explain the observed changes and increased aggressiveness of Pam compared to other pathogens. There are a large number of pathogenicity-associated genes, but only a fraction of them have a known function and have been associated with different hosts. The problem lies when horizontal gene transfer occurs: different *Pseudomonas* species can transfer pathogenicity-related genes to other *Pseudomonas* [12], mostly because of the coexistence between isolates in the same host, especially the bacterial canker-associated isolates. Therefore, the selection of pathogenicity-related genes for primer design must consider bacterial isolates within and outside the respective phylogroup of the target bacteria.

In addition to classical PCR, other methods such as LAMP (loop-mediated isothermal amplification) have been developed with the aim of improving specificity and sensitivity compared to PCR. Specificity and sensitivity are fundamental parameters associated with any detection technique. Comparison between different techniques allows the selection of the best method performed under specific conditions. The most commonly used routine method today is PCR, but nowadays there are other detection methods such as LAMP, which is specifically used for the detection of plant pathogenic bacterial genera, which is the case of *Pectobacterium* [16], *Pantoea* [17], *Xanthomonas* [18], *Pseudomonas* [19,20,21,22,23], and other pathogens such as *Venturia carpophila* [24], *Bipolaris oryzae* [25], or the set of phytoplasmas associated with sesame-phyllody disease [26]. Additional features such as low equipment requirements considering that it could be performed without a PCR machine, amplification under isothermal conditions, and the use of 4 to 6 primers targeting more genomic regions ensure high operational efficiency and are highly specific even against the co-presence of non-target DNA [27]. LAMP can be implemented in both laboratory and field conditions and could be a great alternative to develop detection analysis for this bacterium, especially considering the geographical limitations of Chile and other countries that also have similar conditions. These characteristics make it an attractive method for detection, especially for field development.

The current change in agroclimatic conditions could cause cherry production to migrate to the south in the future, where climatic conditions are more favorable for the development of the disease. Thus, the early detection of Pam represents a fundamental pillar to prevent its spread, limiting the damage that this bacterium could cause in the future to the production of cherries, considering the relevant role that the cherry market plays in the Chilean fruit industry. Therefore, the objective of this work was to optimize the detection of Pam to achieve higher specificity and sensitivity, reduce the cost of analysis and identify any potential genetic variation among strains. To this end, we propose to develop the LAMP technique (loop-mediated isothermal amplification) and a new PCR protocol for the detection of *Pseudomonas amygdali* pv. *morsprunorum*, using a genomic approach.

## 2. Results

### 2.1. Bacterial Isolates Identification

Several sequential steps were carried out to develop a LAMP-based Pam detection protocol. Bacteria isolated from symptomatic *Prunus avium* trees in four geographical regions of Chile were studied, and the DNA of a selected group of twelve strains was sequenced by PCR product sequencing (MLSA) and draft genome sequencing. To study the relationships among the selected bacterial isolates, a phylogenetic tree based on MLSA was constructed using the PCR product sequence of four genes (*cts*, *gyrB*, *pgi*, and *rpoD*) for the 12 isolates sequenced in this study (Table 1) and the available sequence for the same genes from 7 bacterial genomes (PSSB728, PSSB31D, *P. avellanae*, *P. savastanoi*, *P. amygdali* pv. *tabaci*, CFBP2116, and *P. viridiflava*) (Appendix A), representing the phylogenomic species of *Pseudomonas* proposed by Gomila et al. [9]. The phylogenetic tree in Figure 1 shows a cluster formed by the Pam isolates S1 Pam, S2 Pam, 11116B2, and CFBP2116. In addition, the analysis allows us to conclude that the Chilean isolates of Pam are distantly related to Pss isolates (PSSB31D and PSSB728) and to other bacteria from phylogroups I, II, and V.

To complete these results, we performed the ANIb test (Figure 2), comparing the full genome of the same isolates used in the MLSA analysis (CFBP2116) together with reference strains (Appendix A) of other species belonging to phylogroups related to Pam, according to [10]. ANIb results confirm that the isolates S1 Pam, S2 Pam, and 11116B2 correspond to *Pseudomonas amygdali* pv. *morsprunorum* and allowed us to confirm the identification of the other Chilean isolates used in this study, described in Table 1. Further ANIb comparisons were performed in order to establish the genetic relationship of Chilean isolates of Pam with other Pam isolates sequenced worldwide, showing values over 98.7% of average nucleotide identity in comparison with 16 isolates available in genbank (Appendix A).

### 2.2. PCR and LAMP Primer Design for Pseudomonas amygdali pv. morsprunorum

To find unique genome regions between the strains, we used the RAST tool “Sequence based comparison” in order to perform multiple genomic comparisons against CFBP2116 reference genomic sequence. Each comparison was performed simultaneously against 4 genomic sequences of *Pseudomonas* obtained in this work (Table 1) and also against genomic sequences obtained from Genbank (Appendix A), resulting in the selection of the Type III effector hopAU1 gene for the primer design (Table 2). To verify the conservation of the genomic region selected for the design of the primers, we extracted and aligned HopAU1 genomic regions from all sixteen isolates from genbank, showing a complete match of al the primers used for this study (Appendix A)

### 2.3. Primer Validation

First, the primers were tested using the 12 local genomes, indicated in Table 1, to determine their specificity in this group. Regarding PCR primer, there was specific amplification only in the three samples previously identified as Pam, i.e., 11116B2, S1 Pam, and S2 Pam. On other hand, LAMP primers show the same specificity as PCR primers (Figure 3), amplifying only the 3 isolates previously identified as *Pseudomonas amygdali* pv. *morsprunorum*.

### 2.4. Specificity Tests

To determine the specificity of the primers designed, 131 bacterial isolates were collected in fields where Pam was previously detected. The isolates were identified by the amplification and sequencing of 16S rRNA gene and are listed in Appendix A. PCR and LAMP protocols were tested against all the isolates, showing no positive amplification from both primers set. Among those 131 isolates, 30 isolates belonged to *Pseudomonas* genus, which allows us to conclude that both sets of primers are highly specific for Pam. In addition, we can reinforce the idea that the primers are not amplifying any different *Pseudomonas* species besides Pam (Appendix A).

### 2.5. Sensitivity Tests from Pure Bacterial DNA Extracts

Both quality and quantity of a DNA template can dramatically affect the results of each method. To determine sensitivity value of the LAMP and PCR assay, a sensitivity test was performed using one of the positive isolates found for Pam (11116B2). The results showed that the detection limit for PCR amplification was 100 pg/μL, while the detection limit for the LAMP method was at 1 fg/µL (Table 3). The samples used for this test were all purified DNA from selected colonies.

### 2.6. Comparison of Bacterial Detection Techniques from Plant Tissue

PCR and LAMP detection methods were compared in terms of sensitivity, based on the future use of these techniques in the field. The LAMP method has a higher sensitivity than the PCR detection method directly on plant tissue, being able to detect a concentration of 10^3^ ufc/mL of *Pseudomonas amygdali* pv. *morsprunorum* in the solution (Table 4).

## 3. Discussion

This study used two proven bacteria identification tools: ANIb and MLSA. Based on Gomila et al. [9], the use of both techniques simultaneously showed more accurate results in the species-subspecies delimitation. The ANIb test not only helps to identify the three isolates corresponding to Pam, but it can also highlight the differences among these strains, which is a critical point in the development of a detection method. It is expected that a greater difference at the genomic level correlates with the existence of a greater number of unique genes in the Pam isolates, compared to other *Pseudomonas* not necessarily associated with bacterial canker complex. The MLSA showed the same result as ANIb, being the three Chilean isolates identified as Pam (11116B2, S1 Pam, and S2 Pam). However, even when both analyses (MLSA and ANIb) reached similar conclusions, it can be noted that ANIb has better separation and distribution of the different phylogroups in the cladogram, where phylogroup II is less related to phylogroup IV. A larger number of genomes included in both analyses can help to better visualize the differences between the phylogroups.

In terms of sensitivity, LAMP can be a powerful tool. The detection of the pathogen using purified DNA from bacterial culture, at a concentration of 1 fg/µL, implies that Pam can be detected in minimum levels on the tissue sample and achieved similar levels of detection to other LAMP protocols [22]. The detection of *P*. *amygdali* pv. *morsprunorum* direct from plant material with LAMP proves that can it be obtained with low concentration of DNA. This is a good sign for the application of the technique in the field, performing the test even when the bacterial load is minimal in seasons of low microbial activity. To evaluate the minimal concentration of DNA that can be identified in LAMP, it will be necessary in the future to consider lower amounts of bacteria in CFU/mL. In the case of PCR, although it has a lower sensitivity compared to LAMP, this detection method is widely used for agronomical purposes and can be utilized instead of LAMP when there are no equipment/reagents for this purpose. It can be noted that in both detection methods the concentration of DNA is low, and similar to other studies [28,29,30,31].

The specificity of an assay depends on several factors such as the target gene and the number of bacteria, closely or distantly related to the target bacteria, used to define the specificity of the selected gene. Host-associated microbial communities can be highly variable and distinct from those found in the external environment [32,33,34,35]. Therefore, the microbiota associated with cherry trees is a good starting point to increase the number of isolates to validate the specificity of the technique, considering that one of these isolates may cause a non-specific amplification. This, together with the inclusion of non-target bacteria for the same genera, helped to reduce the error in the specificity of a molecular method. With this in mind, for the field samples, spring was considered the best sampling period, where it was expected to find an active microbiota due to the end of dormancy. The relationship between tissue type and microbial community was also considered, taking samples from bark, roots, and phloem, increasing the diversity of bacterial isolates used for specificity tests. We observed several differences even at the same species or genotype level [36], but the method proposed here achieved a high level of specificity within the group of samples tested and in all conditions tested.

Regarding the primers, *hopAU1* was selected as genomic target among 25 isolates of bacterial species related to bacterial canker in cherry trees or associated with other phylogroups belonging to the *Pseudomonas* genera. This search included only part of the isolates that composed the phylogroup IV, and other bacteria belonging to phylogroups I, II, and V. Even though *hopAU1* has been found in other pathovars related to phylogroups II and IV [37], the purpose of these methods was to design a tool that can distinguish bacterial canker and host-associated bacteria in different isolates related to cherry, a goal that was addressed in this work. One of the main considerations was the presence of this gene in some strains of *Pseudomonas syringae* pv. *actinidiae* [38], but the design of the forward primers was made on the promoter region of this gene, which showed higher variability than the coding sequence of the gene. The selection of this gene was promising due to the fact that HopAU1 is an evolutionarily conserved effector in *Pseudomonas* that plays a role in the induction of host cell death [37].

This work presents the first LAMP method based for *Pseudomonas amygdali* pv. *morsprunorum*. In the case of the PCR method, the sensitivity of our protocol is similar to another PCR previously reported in [30]. One of the advantages of this study is that it includes the local microbiota associated with cherry trees and different *Pseudomonas* species associated to bacterial canker. 

The maintenance of quarantine status around Pam for Chile and other countries is a critical step, and one of the most important solutions for this purpose is the improvement of early detection tools.

## 4. Materials and Methods

### 4.1. Strain Isolation and DNA Extraction

The samples for genomic sequencing were collected in the spring of 2020 from four different regions of Chile (Ñuble, Maule, O’Higgins, and Metropolitana), all corresponding to branches with symptoms associated with bacterial canker. The tissue was macerated with 3 mL of distilled water without prior disinfection, from which 20 µL was cultured on Petri plates with King’s B medium (KB) at 28 °C for 24 h. Colonies selected on the basis of morphological criteria and fluorescence in KB media were further grown in 5 mL of LB media at 28 °C for 16 h, and when fully grown, DNA extraction was performed using the GeneJet DNA Purification Kit (Cat. No. K0721, Thermo Fisher Scientific, Waltham, MA, USA) according to the manufacturer’s instructions. 

### 4.2. Bacterial Identification

Selected isolates were analyzed based on PCR 16S rRNA for genera identification and MLSA for four housekeeping genes (*cts*, *gyrB*, *pgi*, and *rpoD*) proposed by Sarkar and Guttman [39]. The 30 μL PCR reaction mixture contains 21 μL of distilled water, 3 μL of buffer 10×, 1.5 μL of MgCl_2_, 1 μL of dNTPs, 0.2 μL of Taq polymerase (Invitrogen^TM^, Waltham, MA, USA), and 1 μL of each primer enlisted, with 1.5 μL of template DNA. The amplification protocol consisted of an initial denaturation at 94 °C for 3 min with 35 cycles as follows: denaturation at 94 °C for 60 s, annealing of primers for 30 s, and elongation at 72 °C for 90 s, with a final extension step at 72 °C for 7 min and then put on 10 °C continuously. The PCR products were separated by electrophoresis in an agarose gel at 1.2%, and positive samples were sent for Sanger sequencing at Psomagen Inc. (Rockville, MD, USA). Whole genomic sequences were obtained using Illumina platform Novaseq6000, with an expected output of 2 Gb using reads of 150 bp paired end. Libraries were constructed with a Truseq nano DNA library prep kit at Psomagen Inc. All the bacterial isolates selected for this study are described in Table 5.

### 4.3. Bioinformatic Analysis and Primer Design

CLC Genomic Workbench (24.0.1) was used for the bioinformatic analysis as isolates identification, draft genome assembly, phylogenetic trees for MLSA, and ANIb analysis. A phylogenetic tree based on the MLSA was built including the sequenced strains and seven selected genome sequences available on GenBank, classified as Pam or part of the *Pseudomonas* phylogroup I, II, IV, and V [9] (Appendix A). The strain CFBP2116, a reference for the *Pseudomonas amygdali* pv. *morsprunorum* species, was included in the ANIb analysis. The selected isolates (Table 1) were submitted to the RAST server 2.0 (https://rast.nmpdr.org/rast.cgi accessed on 3 August 2023) to identify potential genomic regions used for the primer design for PCR and LAMP methods at PrimerExplorer V5 (https://primereBxplorer.jp/lampv5e/ accessed on 21 March 2022).

### 4.4. LAMP Optimization Protocol

The different concentrations of reagents and parameters associated with LAMP protocol were examined. The results showed that the best annealing temperature was 65 °C for 60 min, followed by an incubation at 85 °C for enzyme denaturation. The optimized reaction mixture was adjusted to 1.6 μM of the internal primer, 0.2 μM of the external primers, 6 mM of MgSO_4_, and 1 μL formamide added per sample. Other reagents in the mixture solution stayed at the same concentration.

### 4.5. Detection of Plant Tissues-Derived Pam

During the winter and spring of 2021, samples were collected from thirteen commercial sweet cherry orchards in four regions of Chile, for a total of 145 samples (Table 6). Tissue samples (phloem, bark, and roots) were cultured on KB [40] for bacterial isolation and later identified based on Sanger sequencing for 16Sr RNA gen. A total of 131 bacterial isolates obtained from the tissue sample, classified into 30 genera (Appendix A), were used to compare the detection specificity and sensitivity of each diagnostic method. All the orchards have historical records of bacterial canker, with characteristics symptoms of the disease visible at the time of collection.

### 4.6. Specificity and Sensitivity Tests

The specificity and sensitivity of each technique (PCR and LAMP) were tested, and both methods were compared. In case of specificity, a total of 30 genera of bacteria, which correspond to the 131 isolates obtained in the field sample, were put on test, also including 3 positive controls of Pam and other *Pseudomonas* bacteria enlisted in Table 1. In terms of sensitivity, the DNA of one strain of *Pseudomonas amygdali* pv. *morsprunorum* (11116B2) was adjusted to an initial concentration of 10 ng/µL, from which it was serial diluted to concentrations of 1 ng/µL, 100 pg/µL, 10 pg/µL, 1 pg/µL, 100 fg/µL, 10 fg/µL, 1 fg/µL, 100 ag/µL, 10 ag/µL, and 1 ag/µL. The amplification protocol for PCR analysis consists of an initial denaturation at 94 °C for 3 min followed by 35 cycles with a denaturation process at 94 °C for 45 s, annealing at 65 °C for 30 s, elongation at 72 °C for 45 s, with a last final elongation step at 72 °C for 7 min, lastly with a step at 10 °C continuously, while the 30 μL PCR reaction mixture contains 21 μL of distilled water, 3 μL of buffer 10×, 1.5 μL of MgCl_2_, 1 μL of dNTPs, 0.2 μL of Taq polymerase (Invitrogen^TM^), and 1 μL of each primer enlisted, with 1.5 μL of template DNA. In case of LAMP, 25 μL reaction mixture contains 10.9 μL of distilled water, 2.5 μL of isothermal amplification buffer 10×, 1 μL of MgSO_4_, 3.5 μL of dNTPs, 1.6 μL of each internal primer (FIP and BIP), 0.2 μL of each external primer (F3 and B3), 1 μL of formamide, 1 μL of Bst 2.0 DNA polymerase (New England BioLabs Inc., Ipswich, MA, USA), and 1.5 μL of template DNA. The LAMP protocol consisted of a single step of amplification at 65 °C for 60 min, followed by a temperature increase at 85 °C for 5 min to degrade the enzyme after the amplification process, with a final step at 10 °C to preserve the LAMP products.

### 4.7. Comparison of Detection Techniques from Plant Material

To compare the two techniques, cherry leaves were artificially inoculated with a Pam’s bacterial suspension (11116B2). Bacterial cultures were grown overnight in LB media and diluted to an initial concentration of 107 ufc/mL, then serially diluted tenfold to 103 ufc/mL. Then, 200 µL of the bacterial culture was added to a 0.15 g sample of cherry plant tissue, which was processed in a mortar and pestle. The DNA sample from the leaf tissue plus the bacterial dilution used for this assay was extracted using the modified protocol proposed by Zhang et al. [37]. The volumes and reagents used in this reaction are the same as mentioned in 4.6. The amplification protocol for PCR analysis consisted of an initial denaturation at 94 °C for 3 min, followed by 35 cycles of denaturation at 94 °C for 45 s, annealing at 65 °C for 30 s, elongation at 72 °C for 45 s, with a final elongation step at 72 °C for 7 min, and finally a step at 10 °C continuously. For the LAMP protocol, it was considered a single step of amplification at 65 °C for 60 min, followed by a temperature increase at 85 °C for 5 min to degrade the enzyme after the amplification process, with a final step at 10 °C to preserve the LAMP products. In both protocols, the amplification products were visualized on a 1.2% agarose gel.

## 5. Conclusions

In the present work, the development of LAMP- and PCR-based molecular detection method for *Pseudomonas amygdali* pv. *morsprunorum* race 1 showed satisfactory results, being highly specific and sensitive. LAMP has a better performance and low detection rate, identifying target bacteria at a concentration up to 10^3^ ufc/mL direct from plant tissue and 1 fg/µL in purified DNA from colonies. Either way, both detection methods only detect Pam from a diverse group, consisting of 30 genera, including more than 35 isolates belonging to the genus *Pseudomonas*.

Genome-based molecular tools allow the development of better detection methods, taking into account different sequences, both local and from database, improving the search for unique genes considering a more integrated and larger database.

The novel development of these primers for LAMP and PCR tools helps in the search for quarantine pathogens, strengthening the strategies around crop protection, especially for cherry trees.

## Figures and Tables

**Figure 1 plants-12-04119-f001:**
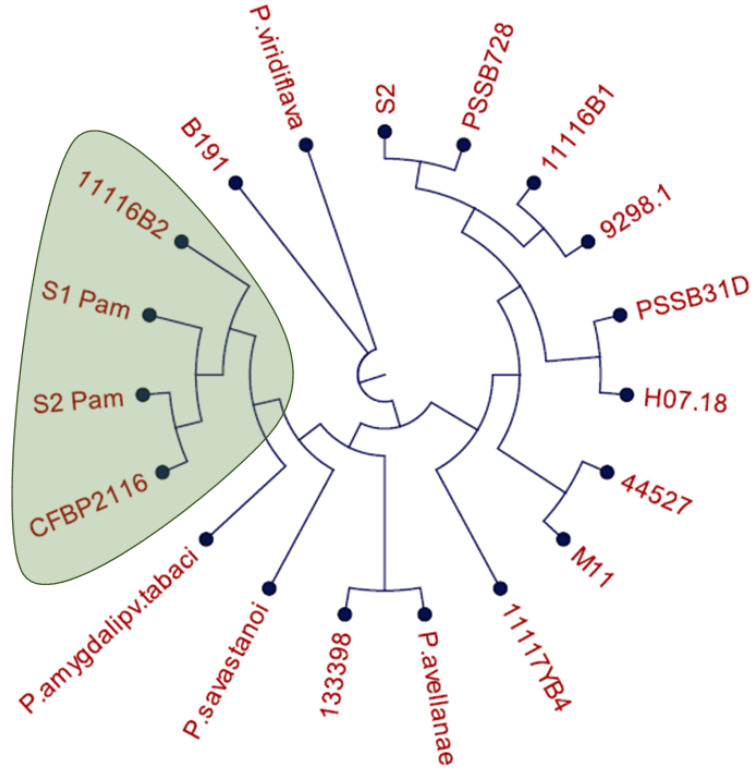
MLSA analysis using 4 housekeeping genes concatenated (*cts*, *gyrB*, *pgi*, and *rpoD*). MLSA includes 7 sequences (Appendix A) obtained from NCBI related to *Pseudomonas* together with 12 local isolates from this study (Table 1).

**Figure 2 plants-12-04119-f002:**
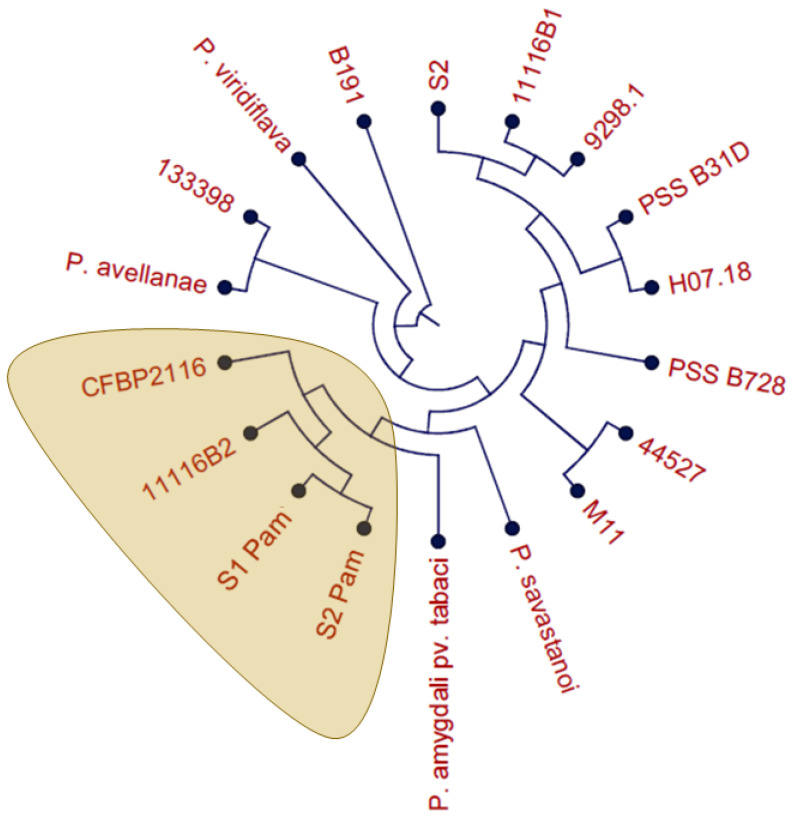
Average Nucleotide Identity (ANIb) calculated tree of the strains used in this study. The local isolates 11116B2, S1 Pam, and S2 Pam cluster together with the reference isolate for Pam, CFBP2116 (Appendix A).

**Figure 3 plants-12-04119-f003:**
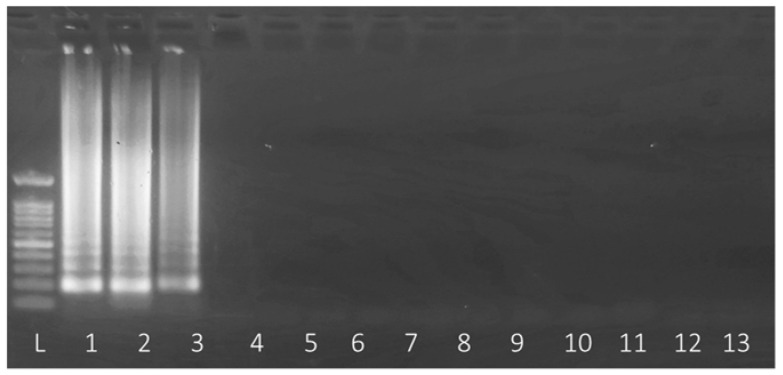
LAMP assay showing the positive amplification of 11116B2 (1), S1 Pam (2), and S2 Pam (3) to HopAU1 primers. Lanes 4 to 12 lanes are, in order, 11116B1, S2, HO7.18, 9298.1, 11117YB4, M11, 44527, 133398, b191, with a negative control (13).

**Table 1 plants-12-04119-t001:** Identification of the 12 local strains identified in this study.

Strain	Species
11116B2	*Pseudomonas amygdali* pv. *morsprunorum*
S1 Pam	*Pseudomonas amygdali* pv. *morsprunorum*
S2 Pam	*Pseudomonas amygdali* pv. *morsprunorum*
133398	*Pseudomonas syringae* pv. *tomato*
44527	*Pseudomonas syringae* pv. unknown
11116B1	*Pseudomonas syringae* pv. *syringae*
9298.1	*Pseudomonas syringae* pv. *syringae*
S2	*Pseudomonas syringae* pv. *syringae*
H07.18	*Pseudomonas syringae* pv. *syringae*
11117YB4 *	*Pseudomonas syringae* pv. *syringae*
M11	*Pseudomonas syringae* pv. unknown
b191	*Pseudomonas paracarnis*

* This strain was identified only by MLSA.

**Table 2 plants-12-04119-t002:** PCR and LAMP primers for *Pseudomonas amygdali* pv. *morsprunorum*.

Detection Method	Primers	Primer Sequence (5′–3′)	Fragment Size (bp)
PCR	HopAU1_F3	GGCCTGAAGCGGCTGAGT	339
	HopAU1_B3	CTGTTTGCGTGATGCCACT	
LAMP	HopAU1_F3	GGCCTGAAGCGGCTGAGT	-
	HopAU1_FIP	TGTTTATTTGACCAGCCGGCAAGAGCTGTCTTTGGAACCCTCCTGTG	
	HopAU1_BIP	AAGCCCGTTCAATCAGTTAGTGCATATTTCATGAGAGCATGACGCTTCT	
	HopAU1_B3	CTGTTTGCGTGATGCCACT	

**Table 3 plants-12-04119-t003:** PCR and LAMP sensitivity test for *Pseudomonas amygdali* pv. *morsprunorum*.

	DNA Concentration		10 ng/µL	1 ng/µL	100 pg/µL	10 pg/µL	1 pg/µL	100 fg/µL	10 fg/µL	1 fg/µL	100 ag/µL	10 ag/µL	1 ag/µL
Diagnostic Method													
PCR		HopAU1 (Pam)	+	+	+	−	−	−	−	−	−	−	−
LAMP		HopAU1 (Pam)	+	+	+	+	+	+	+	+	−	−	−

**Table 4 plants-12-04119-t004:** Sensitivity of LAMP and PCR techniques for detection and identification of *Pseudomonas amygdali* pv. *morsprunorum*. ‘+’ indicates that the amplicon was detected and ‘−’ means that no amplicon was obtained.

	DNA Concentration		10^7^ cfu/mL	10^6^ cfu/mL	10^5^ cfu/mL	10^4^ cfu/mL	10^3^ cfu/mL	Blank (No Bacteria)
Diagnostic Method								
PCR		*Pseudomonas amygdali* pv. *morsprunorum*	+	+	−	−	−	−
LAMP		*Pseudomonas amygdali* pv. *morsprunorum*	+	+	+	+	+	−

**Table 5 plants-12-04119-t005:** Isolates and host identification with each sampling area.

Isolate Code	Hosts	Geographical Location	Ref_Seq
11116B1	*Prunus avium*	San Fernando, R. de O’Higgins	JARNJA000000000
9298.1	*Prunus avium*	San Fernando, R. de O’Higgins	JARNIZ000000000
44527	*Phaseolus vulgaris*	San Fernando, R. de O’Higgins	JARNJB000000000
133398	*Lycopersicum esculentum* P. mil	Quillota, R. de Valparaíso	JASJMY000000000
11116B2	*Prunus avium*	San Fernando, R. de O’Higgins	JASJNB000000000
11117YB4 *	*Prunus avium*	San Fernando, R. de O’Higgins	-
M11	*Actinidia deliciosa*	Chillán, R. de Ñuble	JAROCH000000000
B191	*Prunus avium*	San Fernando, R. de O’Higgins	JASJMX000000000
S2	*Prunus avium*	Curicó, R. del Maule	JARNIX000000000
H07.18	*Prunus avium*	Curicó, R. del Maule	JARNIY000000000
S1 Pam	*Prunus avium*	Osorno, R. de Los Lagos	JASJNA000000000
S2 Pam	*Prunus avium*	Chile Chico, R. de Aysén	JASJMZ000000000

* This strain was sequenced by Sanger sequencing.

**Table 6 plants-12-04119-t006:** Plant material collected from the 4 regions of Chile, enlisted with the corresponding orchards.

Sample Type	Time of Collection	Sampling Area
Branch	Winter 2021	Melipilla, R. Metropolitana
Root	Winter 2021	Melipilla, R. Metropolitana
Branch	Spring 2021	Melipilla, R. Metropolitana
Branch	Spring 2021	Placilla, R. de O’Higgins
		Placilla, R. de O’Higgins
		Placilla, R. de O’Higgins
		Chimbarongo, R. de O’Higgins
Branch	Spring 2021	Romeral, R. del Maule
		Romeral, R. del Maule
		Río Claro, R. del Maule
		Rio Claro, R. del Maule
		Río Claro, R. del Maule
		Molina, R. del Maule
Branch	Spring 2021	San Nicolás, R. del Ñuble
		San Nicolás, R. del Ñuble

## Data Availability

Suggested Data Availability Statements are available in section “MDPI Research Data Policies” at https://www.mdpi.com/ethics (accessed on 2 August 2023).

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
