# Peer review of "Development of a Genome-Informed Protocol for Detection of Pseudomonas amygdali pv. morsprunorum Using LAMP and PCR"

_plants, 2023, doi:10.3390/plants12244119_

Round 1

Reviewer 1 Report

Comments and Suggestions for Authors

In this manuscript the authors describe a genome-based protocol for the detection of Pam using the well-established methods of LAMP and PCR. Overall, the authors follow the right workflow e.g. gene target identification, specificity testing and sensitivity testing but the manuscript lacks robustness and is poorly written with multiple grammar and spelling issues throughout.

I have some major concerns about the results presented by the authors. 

The PCR and LAMP targets only determined based on the analysis of 3 Pam isolates and 7 non-Pam isolates. There are ~20 Pam genome assemblies on NCBI, why weren’t these used? As it stands, I would be concerned that by using only 3 isolates of Pam they would not capture the available genetic diversity.

From the analysis of these 10 isolates, the authors identify HopAU1 as a potential Pam specific target for PCR and LAMP. However, they validate the primers they designed for this target only on these 10 isolates. They may have also tested the specificity on more non-Pam isolates (Supp. Table 2.) but this wasn’t clear. The primers should be tested on more Pam isolates other than the 3 used to design the primers.

Following on from the above point, line178-179 – the authors cannot claim that no Pam isolates were in the 30 Pseudomonas field isolates used to test specificity because they have designed the Pam primers on just 3 isolates. It could be that these primers are not effective at detecting diverse isolates.

HopAU1 has an expected amplicon size of 339bp, but there are multiple non-specific bands in Fig. 3 for both gels and the authors do not label the ladder so product size cannot be determined. The lane IDs for Fig. 3B is also missing– what is lane 4? this looks like it had weak amplification and I presume it is not Pam?

In the methods section the authors do not describe the methods used for WGS sequencing and genome assembly. The The Ref-seq IDs (Table 7) for the Pam isolates and others are not available on NCBI. It  also wasn’t clear how they tested sensitivity of PCR and LAMP in plant material. Was cherry artificial inoculated with Pam (if so, how? Spray inoculation, infiltration) and then gDNA extracted? Or was cherry gDNA extracted and then spiked with Pam gDNA? 

In addition, I have some minor concerns and suggestions: 

In the introduction the authors should explain that is the impact of Pam in cherry in Chile and what are the current diagnostic tools for detection of Pam and why is a molecular method needed?

The authors collect 12 bacterial isolates from symptomatic cherry in 2020 from 5 different regions. Only 3 of these isolates were Pam. It would be good to include geographical location on the trees (Fig. 1 and 2) in addition to Table 7.

Fig.2 would be better as a table of ANI % rather than a tree or a tree with % ANI included.

Line 162, the authors state that the LAMP PCR detects all 3 Pam isolates and refer the reader to Fig. 3. But the LAMP results in Figure 3A only includes 1 of 3 Pam isolates?

In discussion, the second paragraph reads more like introduction rather than discussion. The authors also mention in discussion that hopAU1 is present in other Pseudomonads – it would be good to include this kind of analysis in the results section.

Comments on the Quality of English Language

 Overall, the authors follow the right workflow e.g. gene target identification, specificity testing and sensitivity testing but the manuscript lacks robustness and is poorly written with multiple grammar and spelling issues throughout.

Author Response

Reviewer 1

Comments and Suggestions for Authors

In this manuscript the authors describe a genome-based protocol for the detection of Pam using the well-established methods of LAMP and PCR. Overall, the authors follow the right workflow e.g. gene target identification, specificity testing and sensitivity testing but the manuscript lacks robustness and is poorly written with multiple grammar and spelling issues throughout.

I have some major concerns about the results presented by the authors. 

The PCR and LAMP targets only determined based on the analysis of 3 Pam isolates and 7 non-Pam isolates. There are ~20 Pam genome assemblies on NCBI, why weren’t these used? As it stands, I would be concerned that by using only 3 isolates of Pam they would not capture the available genetic diversity.

Author’s response: The twelve isolates in this experiment, including the three recognized as Pam species, were collected and their genomes were sequenced. This is due to the low prevalence of Pam in Chile, where the disease was only reported in 2019. We tested the new techniques in other Chilean isolates that were not included in the main text because they were collected in the same tree or from the surrounding trees in the orchard, so we assume they were clones (at least 16SrRNA gene was identical), thus, we considered only the genome-sequenced isolates to show the amplifications on the main figures. On the other hand, we obtained 8 isolates of Pss and other 30 Pseudomonas species (all of them obtained in Chile) that were analyzed and sequenced for 16SrRNA gene and were later used for subsequent tests.

In addition, to provide more diversity during the earlier stages of the design and development of these molecular tools, we incorporated Pam genomes obtained from Genbank (such as CFBP2116) to test the effectiveness and specificity of the LAMP and PCR. To justify the use of just one reference, we performed an additional genomic comparison (included after this revision as figure S1) that demonstrates that the minimum ANIb values is 98,73%, suggesting a low diversity among the full collection of Genbank strains. Moreover, we also observed that only 16 of the 20 species listed on Genbank belong to the Pam species, with four of them having an 88% of similarity to the reference strain CFBP2116, being discarded as part of Pam species.

To validate if the primers are not missing any potential genetic variant of Pam, we aligned the LAMP target region of these 16 isolates, highlighting the primers designed in this work and showing that the six LAMP primers matched all the isolates available together with our isolates. This data was included as supplementary figure S2.

From the analysis of these 10 isolates, the authors identify HopAU1 as a potential Pam specific target for PCR and LAMP. However, they validate the primers they designed for this target only on these 10 isolates. They may have also tested the specificity on more non-Pam isolates (Supp. Table 2.) but this wasn’t clear. The primers should be tested on more Pam isolates other than the 3 used to design the primers.

Authors' response:

As we mentioned in the first question, there are no more isolates identified as Pam in Chile, due to its low prevalence which is possibly related with the recent introduction in the country.

As we indicated in the text, the specificity assays were performed including the group of 131 isolates collected in the field and the twelve isolates fully sequenced. We modified the text between the lines 177-183 to make it clearer. In addition, we performed the alignment indicated before as figure S2 to state that the selected target region was appropriate and conserved among Pam isolates known up to date.

Following on from the above point, line 178-179 – the authors cannot claim that no Pam isolates were in the 30 Pseudomonas field isolates used to test specificity because they have designed the Pam primers on just 3 isolates. It could be that these primers are not effective at detecting diverse isolates.

Authors' response:

The field-collected isolates designated as Pseudomonas were all previously identified using 16Sr RNA gene sequencing, so we are sure we do not have other Pam isolates among them (written in line.

HopAU1 has an expected amplicon size of 339bp, but there are multiple non-specific bands in Fig. 3 for both gels and the authors do not label the ladder so product size cannot be determined. The lane IDs for Fig. 3B is also missing– what is lane 4? this looks like it had weak amplification and I presume it is not Pam?

Authors' response:

The gel shown in the cited figure (3) is a LAMP gel, not a PCR gel. LAMP amplification produces a pattern of successive amplification resembling a DNA ladder. To clarify the objective of figure 3, we included a single gel showing the amplification of the three Pam isolates in comparison of the other nine isolates sequenced in this work. In order to show the differences between PCR and LAMP, we added the PCR gel with the 339 bp fragment of the three Pam isolates against all the Pseudomonas isolates used in this work as a supplementary Figure S3.

In the methods section the authors do not describe the methods used for WGS sequencing and genome assembly. The The Ref-seq IDs (Table 7) for the Pam isolates and others are not available on NCBI. It  also wasn’t clear how they tested sensitivity of PCR and LAMP in plant material. Was cherry artificial inoculated with Pam (if so, how? Spray inoculation, infiltration) and then gDNA extracted? Or was cherry gDNA extracted and then spiked with Pam gDNA? 

Authors' response:

We described the WGS sequencing and genome assembly on text between-the-lines 290-293 in materials and methods to address this comment. In addition, we requested the immediate release of the genome sequences to genbank administration.

In the lines 341 to 347, the text was modified in order to provide clarification of the process used to create the artificial inoculation.

In addition, I have some minor concerns and suggestions: 

In the introduction the authors should explain that is the impact of Pam in cherry in Chile and what are the current diagnostic tools for detection of Pam and why is a molecular method needed?

Authors' response: Done.

We modified the text to justify this work based on the suggestions. It is indicated the existence of a previous molecular method and provide the citation for the paper in the discussion section.

The authors collect 12 bacterial isolates from symptomatic cherry in 2020 from 5 different regions. Only 3 of these isolates were Pam. It would be good to include geographical location on the trees (Fig. 1 and 2) in addition to Table 7.

We decided not to modify the tree, to avoid the extensive addition of text in the figure. We consider that is enough to include this as a table.

Fig.2 would be better as a table of ANI % rather than a tree or a tree with % ANI included.

Authors' response: Done.

We maintained the figure and added a new figure in the Supplementary material (Suppl. Figure S1) of the ANIb percentages.

Line 162, the authors state that the LAMP PCR detects all 3 Pam isolates and refer the reader to Fig. 3. But the LAMP results in Figure 3A only includes 1 of 3 Pam isolates?

Authors' response:

In order to clarify this images, we decided to include just one new image including the three Pam strains amplified by Lamp, in comparison with the remaining 9 strains used for validation of the primers, and labeled as Figure 3. For PCR validation, we included a new supplementary Figure S3, showing specific amplification of Pam isolates in comparison with 30 Pseudomonas sp. isolates, 8  Agrobacterium sp isolates and one Xanthomonas sp. isolate.

In discussion, the second paragraph reads more like introduction rather than discussion. The authors also mention in discussion that HopAU1 is present in other Pseudomonads – it would be good to include this kind of analysis in the results section.

We modified the discussion and introduction according to reviewers suggestions.

Comments on the Quality of English Language

Overall, the authors follow the right workflow e.g. gene target identification, specificity testing and sensitivity testing but the manuscript lacks robustness and is poorly written with multiple grammar and spelling issues throughout.

Authors' response:

The required change was made regarding English grammar and spelling.

Reviewer 2 Report

Comments and Suggestions for Authors

The reviewed manuscript is dedicated to the design and validation of droplet digital PCR assay detecting Pseudomonas amygdali pv. morsprunorum (Pam), a dangerous pathogen of plants. The presented results are interesting for scientists, specializing on the field of molecular diagnostics. However, a number of issues needs to be addressed before publication.

Major issues:

1.      Page 2, lines 55-86. Authors are encouraged to partly transfer this paragraph in the Discussion section and to rewrite the Introduction. Specifically, more epidemiological context of Pam together with phylogeny of the Pseudomonas genus and diagnostics methods for its detection would be highly appreciated and increase the readability of the manuscript.

2.      It is known that loop primers accelerate the speed of LAMP reaction. Authors are encouraged to explain, why such primers were not designed. Also, the alignment with targeted amplicons would demonstrate how conservative are the selected primers.

3.      2.3. Primer validation — the results of the PCR with different strains need to be added. Detailed protocols for initial primer’s validation are not provided in the Material and Methods section.

4.      Figure 3 — what were DNA samples on the B panel? Also, lane 4 seems to be positive too.

5.      2.5. Sensitivity tests from pure bacterial DNA extracts and 2.6. Comparison of bacterial detection techniques from plant tissue — authors are encouraged to explain the difference between PCR and LAMP sensitivity. Were all Pseudomonas isolates used in the experiment different from Pam?

6.      The protocol for PCR need to be specified.

7.      What DNA was used as a positive control in initial experiments with PCR and LAMP?

8.      4.5. Detection of plant tissues-derived Pam — what was the amount of samples type and were they taken from different of same trees?

Minor issues:

1.      Language needs to be polished in terms of the style.

2.      2.1. Bacterial isolates identification — the sequenced genes need to be listed.

3.      Table 2 — the length between F3 and B3 needs to be added as the size of the LAMP fragment.

4.      4.2. Bacteria identification — the used kit for Sanger sequencing needs to be specified, as well as concentrations and sequences of primers. Also, details of NGS protocol are necessary too.

5.      4.4. LAMP optimization protocol — ranges of primers concentration, reaction temperature, the source of LAMP reagents, DNA concentration are necessary.

6.      4.6. Specificity and sensitivity tests — what was the DNA concentration in the specificity test?

7.      Page 12, line 359 “increase in temperature at 85°C for 5 minutes” — plausibly, this step was intended for inactivation of an enzyme in LAMP.

Comments on the Quality of English Language

Language needs to be polished in terms of the style.

Author Response

Reviewer 2

Comments and Suggestions for Authors

The reviewed manuscript is dedicated to the design and validation of droplet digital PCR assay???? detecting Pseudomonas amygdali pv. morsprunorum (Pam), a dangerous pathogen of plants. The presented results are interesting for scientists, specializing on the field of molecular diagnostics. However, a number of issues needs to be addressed before publication.

Authors' response: Is important to note that the paper is not about droplet digital PCR assay.

Major issues:

  1. Page 2, lines 55-86. Authors are encouraged to partly transfer this paragraph in the Discussion section and to rewrite the Introduction. Specifically, more epidemiological context of Pam together with phylogeny of the Pseudomonasgenus and diagnostics methods for its detection would be highly appreciated and increase the readability of the manuscript.

Authors' response:

To make the introduction more readable without sacrificing coherence, we adjusted the text and some of the content you mentioned was moved to the discussion section.

  1. It is known that loop primers accelerate the speed of LAMP reaction. Authors are encouraged to explain, why such primers were not designed. Also, the alignment with targeted amplicons would demonstrate how conservative are the selected primers.

Authors' response:

Regarding the point of why we don't include the loop primers, it is true that doing so could speed up the reaction, but not all primers designed for LAMP protocols require the addition of the loop primers. We cited a paper by Shirshikov and Bespyatykh (2022) as a source:

“However, it should be taken into account that the probability of detecting annealing sites of loop primers directly depends on the length of the corresponding loops of the starting structures. For some sets of “core” primers, which is a combination of inner and outer primers [23], only one loop primer can be designed, while for others, none. It is for this reason that not every test system can be supplemented with loop primers”.

We put the four primers we designed for this work to the test, and when compared to other published methods, their functionality and amplification time rate were average.

  1. 2.3. Primer validation — the results of the PCR with different strains need to be added. Detailed protocols for initial primer’s validation are not provided in the Material and Methods section.

Authors' response:

The image of the PCR amplification including the 12 (Table 1) and 30 strains (Suppl. table 2) of Pseudomonas was added as a Supplementary material. We added the protocol in section 4.6 of materials and methods.

  1. Figure 3 — what were DNA samples on the B panel? Also, lane 4 seems to be positive too.

Authors' response:

To avoid confusing the reader, we modified the image, and we rewrite the text to make it clearer.

  1. 2.5. Sensitivity tests from pure bacterial DNA extracts and 2.6. Comparison of bacterial detection techniques from plant tissue — authors are encouraged to explain the difference between PCR and LAMP sensitivity. Were all Pseudomonas isolates used in the experiment different from Pam?

Authors' response:

We added a new paragraph to the discussion section in which we explain why we believe there are distinctions between the two processes. Due to the limited incidence of this species in Chile, the results demonstrate that all the other Pseudomonas were distinct from Pam and were validated by 16SrRNa gene amplification and sequencing.

  1. The protocol for PCR need to be specified.

Authors' response:

      We added the protocols in Material and methods section. In the case of plant material extraction, the amplification protocol can be found in section 4.7 of materials and methods.

  1. What DNA was used as a positive control in initial experiments with PCR and LAMP?

Authors' response:

As we stated on the paper, Pam was considered a quarantine bacterium in our country, thus, our positive controls were the same isolates we collected and sequenced completely. With this sequencing validation, we are sure we use Pam for the design and validation of the primers. To show the specificity of the primer design, we included a new figure as supplementary material where it is shown an alignment of the target region extracted of all known isolates of Pam worldwide.

  1. 4.5. Detection of plant tissues-derived Pam — what was the amount of samples type and were they taken from different of same trees?

Authors' response: Done

We added this explanation in the text in the 4.7 section of materials and methods.

Minor issues:

  1. Language needs to be polished in terms of the style.

Authors' response: Done

  1. 2.1. Bacterial isolates identification — the sequenced genes need to be listed.

Authors' response: Done

  1. Table 2 — the length between F3 and B3 needs to be added as the size of the LAMP fragment.

Authors' response:

The length between F3 and B# is included in the table. The size of the LAMP fragment depends is not applicable because the amplification depends on the initial size of F3 and B3, thus, is not necessary to include this in the Table.

  1. 4.2. Bacteria identification — the used kit for Sanger sequencing needs to be specified, as well as concentrations and sequences of primers. Also, details of NGS protocol are necessary too.

Authors' response: Done

  1. 4.4. LAMP optimization protocol — ranges of primers concentration, reaction temperature, the source of LAMP reagents, DNA concentration are necessary.

Authors' response: Done

  1. 4.6. Specificity and sensitivity tests — what was the DNA concentration in the specificity test?

Authors' response: Specified in the text

  1. Page 12, line 359 “increase in temperature at 85°C for 5 minutes” — plausibly, this step was intended for inactivation of an enzyme in LAMP.

Authors' response:

Corrected in the main text.

Comments on the Quality of English Language needs to be polished in terms of the style.

Authors' response: Done

Round 2

Reviewer 2 Report

Comments and Suggestions for Authors

Many thanks to authors for their careful explanations and editing the manuscript. All questions were answered making the article clearer and more readable. However, there still are minor issues, meaning occasional typos and not italicized species names, absence of PCR details (reagent’s manufacturer, reaction volume, volume or amount of template, primer’s concentrations).

Author Response

Reviewer 2

Many thanks to authors for their careful explanations and editing the manuscript. All questions were answered making the article clearer and more readable. However, there still are minor issues, meaning occasional typos and not italicized species names, absence of PCR details (reagent’s manufacturer, reaction volume, volume or amount of template, primer’s concentrations).

Authors' response: Done